# Effectiveness of mindfulness-based online therapy or internet-delivered cognitive behavioral therapy compared with treatment as usual among patients with persistent somatic symptoms: Protocol for a randomized controlled trial

Velina Vangelova-Korpinen[1]*, Helena Liira[1], Samu N. Kurki[2,3‡], Markku Sainio[1], Antti Malmivaara[4,5,6], Mari Kanerva[1,7], Jan-Henry Stenberg[8‡], Mikko Varonen[1‡], Mikko Venäläinen[9‡], Piia Vuorela[10], Jari Arokoski[11]

**1** Outpatient Clinic for Persistent Symptom Rehabilitation, Helsinki University Hospital and University of Helsinki, Helsinki, Finland, **2** Faculty of Biological and Environmental Sciences, Molecular and Integrative Biosciences, University of Helsinki, Helsinki, Finland, **3** Neuroscience Center, HiLIFE, University of Helsinki, Helsinki, Finland, **4** Finnish Institute for Health and Welfare, Helsinki, Finland, **5** Orton Orthopaedic Hospital, Helsinki, Finland, **6** University of Helsinki, Helsinki, Finland, **7** Infection Control Unit, Turku University Hospital, The Wellbeing Services County of Southwestern Finland, Turku, Finland, **8** Department of Psychiatry, Brain Center, Helsinki University Hospital and University of Helsinki, Helsinki, Finland, **9** Department of Medical Physics, Turku University Hospital, Turku, Finland, **10** Finnish Medicines Agency Fimea, Helsinki, Finland, **11** Division of Rehabilitation, Department of Internal Medicine and Rehabilitation, Helsinki University Hospital and University of Helsinki, Helsinki, Finland

☯ These authors contributed equally to this work.
‡ SNK, JHS, MV and MV also contributed equally to this work.
* velina.vangelova-korpinen@hus.fi

## Abstract

### Background

Persistent somatic symptoms unexplained by a defined medical or psychiatric condition often include a component of central sensitization. Many treatment options are based on cognitive behavioral therapy. Effective widely available therapies are scarce. There are self-management programs and e-therapies that aim at overcoming the central sensitization by modifying interoceptive neural networks in the brain.

### Objectives

This study aims to investigate the effect of a mindfulness-based amygdala and insula retraining (AIR) online program and an internet delivered therapist assisted therapy offered by Helsinki University Hospital (HUS iCBTpss) compared to treatment as usual (TAU) in the treatment of conditions causing persistent somatic symptoms.

**Data Availability Statement:** Data sharing is not applicable to this article as no datasets were generated or analyzed during the current study.

**Funding:** This work is supported by European Union's Horizon Europe research and innovation programme (Long Covid HEU Grant no. 101057553 2022-2026) and State Research Funds. Open access funded by Helsinki University Library. The funding sources have no other role in the study.

**Competing interests:** Helena Liira and Markku Sainio have received honorariums for lectures and workshops about functional disorders and post-Covid-19 condition. Jan-Henry Stenberg has received lecture fees from Alma Media, Biogen, Boehringer-Ingelheim, Janssen, Lilly, Lundbeck, Merck, Otsuka, Professio, Sanofi Genzyme. Helena Liira, Markku Sainio and Velina Vangelova-Korpinen are members of the Oslo Chronic Fatigue Network. The remaining authors declare no conflicts of interest relating to this manuscript.

## Methods

We will perform a randomized controlled trial aiming at 360 patients. Consenting patients will be randomized to three study arms: online AIR program, HUS iCBTpss (both interventions as add-ons to TAU); and TAU. Functional ability and quality of life surveys will be collected from participants at baseline and at 3, 6, and 12 months after entering the study.

## Conclusions

This study is one of the first to explore non-drug based online interventions developed to overcome the brain's central sensitization and are available and accessible to patients both in primary and secondary care. The results will develop the management of the common, often debilitating persistent somatic symptoms related to many conditions unexplained by a defined somatic or psychiatric illness.

## Trial registration number

NCT05212467.

## Introduction

Persistent somatic symptoms (PSS) is an umbrella term to describe subjectively distressing somatic complaints, irrespective of their aetiology, that are present on most days for at least several months [1]. Often the term is used to refer to physical symptoms that cannot be attributed to any known physical or psychiatric disorder [2]. These conditions have also long been known as medically unexplained symptoms [3]. Mounting evidence shows that central nervous system and brain-body interaction has a significant role in these conditions, and recently they have been referred to as central sensitization syndromes [4].

Most likely, the origin of persistent somatic symptoms is multifactorial and includes an interaction of biological, psychological, and social factors. There is often predisposing vulnerability related to genetic or developed traits and attitudes that triggers and maintains symptoms in these conditions [5]. Usually, the onset is a consequence of stress, psychological trauma, or a stressful biological event such as a viral infection. These predisposing, triggering and maintaining factors lead to structural, functional and chemical changes in the central nervous system causing alterations in processing of internal and external stimuli, i.e. central sensitization [6]. The mechanisms of central sensitization have been widely studied among patients with chronic pain [4, 6, 7].

Chronic fatigue syndrome (ME/CFS) has been studied for at least four decades but no specific pathology or biomarker has been identified so far [8, 9]. A novel challenge causing persistent somatic symptoms is also post COVID-19 condition. It is characterized by debilitating symptoms (including breathlessness, chest pain, palpitations, and fatigue) which can last for months after the acute illness [10, 11]. The symptoms of post COVID-19 condition resemble those of other post-viral conditions, including chronic fatigue syndrome [12]. So far there is no clear explanation for the full range of symptoms in post COVID-19 condition. Researchers have suggested a broader causal model of interaction of biological, social, experiential, and psychological factors related to the condition [13]. Fibromyalgia is characterized by widespread musculoskeletal pain often associated with fatigue, sleep disturbances and other cognitive and somatic symptoms. The pathophysiology of fibromyalgia is not completely understood but

abnormal central pain processing is considered as the primary pathophysiologic mechanism. Genetic factors, female sex and other painful conditions are main predisposing factors [14]. A novel study has shown association between childhood adverse events and fibromyalgia [15].

Because of the multi-factorial origin of conditions with persistent somatic symptoms, treating these disorders is considered challenging by physicians. Simultaneously these are common condition and for example in Denmark it has been estimated that 16% of the adult population suffer from them [16]. Clinics for functional disorders, psychosomatics or central sensitization syndromes are opened in many countries to develop the care of these disorders. At Helsinki University Hospital (HUS) such a clinic was opened in 2019.

Many treatment options are based on cognitive behavioral therapy (CBT). A review of randomized controlled trials assessing the efficacy of treatment for somatoform disorders indicated that CBT is the best-established treatment for a variety of these disorders [17]. Later, a Cochrane review assessed psychological therapies, especially CBT, in medically unexplained symptoms [18]. They were superior to treatment as usual or waiting list in terms of reduction of symptom severity, but effect sizes were small. Similarly, another Cochrane review found that CBT is effective in reducing the symptoms of chronic fatigue syndrome (ME/CFS) compared with usual care and may be more effective in reducing fatigue symptoms compared with other psychological therapies [19].

However, CBT is not widely available and patients suffering from persistent somatic symptoms are many. There is some evidence that CBT administered as internet therapy is effective in ME/CFS [20]. Acceptance and commitment therapy (ACT) is a form of cognitive behavioral therapy which uses acceptance and mindfulness strategies together with commitment and behavior-change strategies to strengthen behavior consistent with personal values as well as to increase psychological flexibility [21]. There is evidence that ACT in a Web-based delivery format was effective in the management of depression [22]. A web-based ACT-program has been developed by the University of Jyväskylä with expertise from the Finnish Institute of Occupational Health and is provided by the Hospital District of Helsinki and Uusimaa, which has already provided web-based treatments for various other conditions.

Easily available non-drug-based therapies are needed for the treatment of patients with persistent somatic symptoms. There is some evidence that neuroplasticity techniques may be effective in the treatment of ME/CFS [23, 24] and fibromyalgia [25] as well as post COVID-19 condition [26]. The amygdala and insula retraining program is developed by Ashok Gupta and is based on the hypothesis, that in the presence of dysregulating factors (e.g. physiological stressors), the amygdala can become chronically sensitized to bodily signals by a traumatic event (e.g. viral infection) leading to a chronic over-activation of the sympathetic nervous system [24].

The insular cortex (insula) is located deep within the lateral sulcus of the brain, and a large body of evidence points to its role as the principal hub for integrating afferent information of the internal state of the body (i.e., interoception) [27–29]. The insula has an antero-posterior gradient of functional organization. The posterior insular regions receive ascending inputs from the autonomic nervous system as well as the spinal cord and brainstem nuclei, thus generating a primary interoceptive representation of the body [27, 30]. The anterior insula in turn connects to multiple other cortical regions and integrates the visceral information into emotional, cognitive and motivational conditions as well as attentional mechanisms to form a comprehensive state of awareness [30–32]. Importantly, such an interoceptive network enables conscious processing of visceral events as evidenced by functional imaging demonstrating evoked insular activity during heartrate detection tasks [33].

Furthermore, the insular interoceptive network is critically involved in sensing the state of the immune system and encoding immune responses [34, 35], but also generating the core

symptoms of sickness and reorienting basic motivational states in response to peripheral inflammatory stimulus such as infection [36]. Recent evidence from experimental animals has also strikingly suggested that the insular cortex can control the peripheral immune system and ignite immune responses similar to previously experienced infections by retrieving respective immune memories (i.e., immune conditioning), which provides a hypothetical mechanism for psychogenic somatic symptoms [35].

Amygdala is a part of the subcortical limbic system with a central role in fear, threat detection and affective processing [37, 38]. It has been repeatedly linked with depressive symptoms and anxiety-related behaviors in the context of inflammation and acute sickness [39–41]. Notably, the amygdala is strongly interconnected with the insular cortex and actively involved in the processing of visceral afferent signals and the physiological state of the body [27, 36]. Meditation training has been shown to decrease the reactivity of the amygdala to emotional stimuli also during non-meditative states [42].

Given the above evidence, it is highly intriguing that behavioral interventions such as mindfulness have been demonstrated to significantly increase the mean connection strength of the insula-centered interoceptive network, suggesting that this system can be engaged and modulated by such low-cost approaches [43].

Overall, there is strong evidence of the structural and functional changes in the brain following mindfulness meditation in numerous conditions, including chronic pain, a widely studied condition of central sensitization [44–49].

The above findings set the theoretical basis for this study which will compare two online interventions: (1) mindfulness-based online amygdala and insula retraining (AIR) program, and (2) ACT based HUS internet therapy developed for persistent somatic symptoms (HUS iCBTpss), and (3) treatment as usual, in conditions causing persistent somatic symptoms (a.o. fibromyalgia, chronic fatigue syndrome (ME/CFS) and post COVID-19 condition). The study will be a randomized controlled trial performed in two clinics at Helsinki University Hospital (The Outpatient Clinic for Functional Disorders and The Outpatient Clinic for Long-term Effects of COVID-19). The trial will clarify whether internet-based non-drug therapies are helpful in overcoming the assumed central sensitization and ensuing functional disabilities caused by these conditions.

In this study we will include patients suffering from persistent somatic symptoms according to the above definition with known physical or psychiatric disorders as causes for the symptoms excluded. In addition, we will apply the specific diagnostic criteria of fibromyalgia and chronic fatigue syndrome, as well as post COVID-19 condition [10, 11, 14], all of which have elements of a dysfunctional autonomous nervous system.

We aim to explore the effect of the AIR program and the HUS iCBTpss on the patients' self-assessment of functional ability and quality of life as compared to treatment as usual. The first hypothesis is that the AIR program is equal to the internet therapy in the management of persistent symptoms observed in bodily stress syndromes, fibromyalgia, fatigue syndrome and post COVID-19 condition. Another hypothesis is that both the AIR program and the internet therapy are more effective than treatment as usual.

## Methods

### Design and setting of the study

The study is a multi-center randomized controlled trial exploring the effects of the mindfulness-based amygdala and insula retraining program and the HUS iCBTpss internet therapy on patient reported functional ability and quality of life in comparison to treatment as usual. We will aim at recruiting 360 volunteering patients. The SPIRIT Schedule of enrollment,

interventions and assessments is shown in Fig 1. The trial design is presented in Fig 2. The protocol follows the SPIRIT guidelines (S1 Checklist. SPIRIT 2013 Checklist).

## Study flow

In the beginning, the patient receives written information and an oral explanation of the study from the physician and is asked to give written informed consent. Patients have the right to withdraw from the study at any point if they so wish with no consequences to their treatment.

The patients are asked to give their consent to participate in the study via the Suomi.fi-service, an electronic service requiring strong identification. The research assistant signs the consenting patients in the Helsinki University Hospital Electronic Case Report Form (HUSeCRF) system which randomizes the patients. The patients then fill in the baseline information forms online in the HUSeCRF system. The research assistant has access to the interim results. In case the research assistant reports significant differences in outcomes between intervention and control groups midway through the study, the study will be terminated earlier than planned.

## Follow-up

The follow up is primarily performed with the HUSeCRF system. The patient receives reminders by text messages and emails when it is time to fill in the 3, 6 and 12 month follow up surveys. If a patient has not filled in the questionnaires after three reminders, the study nurse contacts him or her by telephone to collect the outcome data.

## Study duration and timeline

The study will have an overall duration of approximately 30 months and will be considered completed at the latest 12 months after the recruitment target is achieved.

The study timeline is the following:

Months 1–24: Selection and enrolment (starting from 1.1.2022) of the patients and database implementation.

Months 3–36: Follow-up (3, 6, and 12 months) and database implementation.

Months 24–36: Analysis of data and final dissemination of results.

## Participants and recruitment

**Inclusion criteria.**   Age 18 to 65 yearsPersistent somatic symptoms, fibromyalgia, chronic fatigue syndrome, and post COVID-19 as predefined subgroups (please see S1 Appendix. Diagnostic criteria).

- Diagnostic examinations have ruled out the potential somatic or psychiatric reasons for the symptoms AND

- Disabling symptoms have lasted at least 3 months AND

- Patient is willing to receive an online, mostly self-management intervention.

**Exclusion criteria.**

- Patients for whom participation could be overly demanding because of physical constraints (for example, patients who cannot write because of dystonia or bedridden patients)

- Patients with presence of severe psychiatric and severe somatic disorders (e.g. moderate or severe depression or newly onset cancer) for whom the study could be overly strenuous.

| | Study period | | | | | |
|---|---|---|---|---|---|---|
| | Enrolment | Allocation | Post-Allocation | | | Close-out |
| Timepoint | $-t_1$ | 0 | 0 months | 3 months | **6 months** | 12 months |
| **ENROLMENT** | | | | | | |
| Eligibility screen | X | | | | | |
| Informed consent via suomi.fi | X | | | | | |
| Randomization and Allocation | | X | | | | |
| **INTERVENTIONS** | | | | | | |
| (TAU +) AIR-trial (duration c. 10 weeks) | | | | | | |
| (TAU+) HUS iCBTPss (duration c. 10 weeks) | | | | | | |
| **CONTROL** | | | | | | |
| TAU (6 months) | | | | | | |
| **ASSESSMENT** | | | | | | |
| **Baseline variables:** age, gender, diagnosis, comorbidity, social status, occupation | X | | | | | |
| **Primary outcomes:** -functional ability (0-10) - quality of life (0-10) - WHODAS2.0 | | | X | X | **X** | X |
| **Secondary outcomes:** -PREMs -EuroHIS-QOL-8 -15D -SSD-12, PHQ-15 -PHQ-9 -GAD-7 -ISI -work ability (1-3) - RS-14 | | | X | X | **X** | X |
| **Other:** -diagnoses, medication, healthcare use, other treatments during 12 months | | | | | | X |

**Fig 1. SPIRIT schedule.** Schedule of enrollment, interventions and assessments.

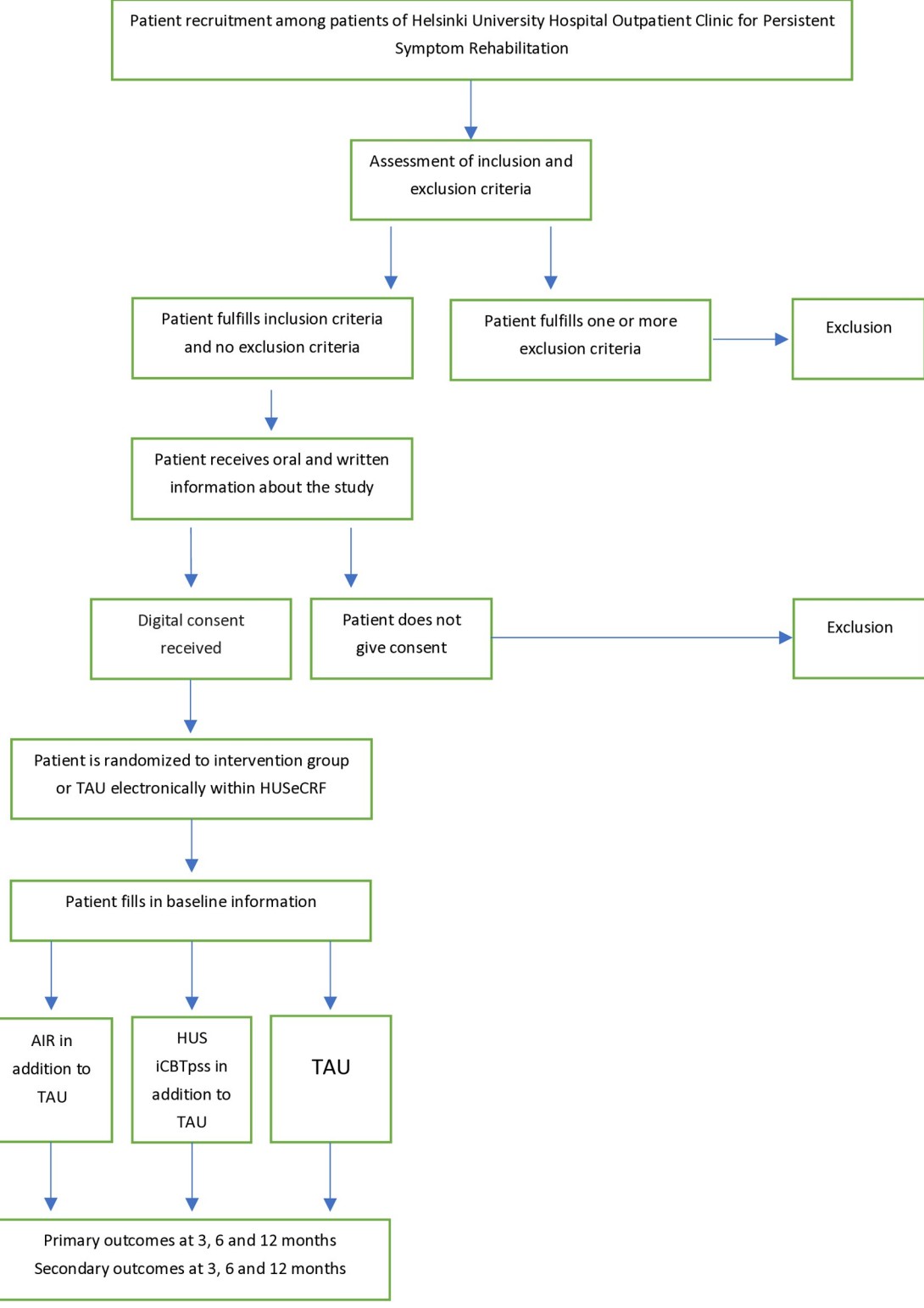

**Fig 2. Study design.**

• Patient has received either or both of the investigated interventions previously

## Interventions

**Amygdala and insula retraining program (AIR).**   The amygdala and insula retraining program (AIR) consists of novel brain retraining approaches focused on hypothetically interrupting and retraining the conditioned defensive hyper-stimulation of the sympathetic nervous system and aspects of the immune system by the amygdala and insula, to bring the brain and body back to homeostasis [23]. It includes supportive techniques such as breathing, meditation, and neurolinguistic programming. The patients are invited to an online program of video session and eight weekly two hour webinar sessions followed by three monthly webinar sessions. The patients are assigned to do daily homework that takes approximately 15 to 20 minutes to complete. The provider of the program monitors the participation of the patients in the program. The patient needs to have an e-mail address, an electronic device (smartphone, tablet or computer) and means for electronic identification to receive the intervention. The AIR program is offered to patients of the outpatient policlinics only as a part of this randomized controlled trial.

In addition to the study intervention, patients receive treatment as usual (TAU).

**HUS internet delivered therapist assisted therapy for persistent somatic symptoms.** The HUS internet delivered therapist assisted therapy for persistent somatic symptoms is based on acceptance and commitments therapy (ACT) and includes psychoeducation as well as exercises aimed at improving wellbeing by relaxing the body and learning new ways of observing the symptoms, and psychological flexibility by developing acceptance and self-compassion. The program consists of ten modules at 1-week intervals lasting 10 weeks altogether. The participant receives weekly written feedback on each module by a trained therapist (psychologist, nurse or social worker). The provider of the program monitors the steps finished and the progress made by the patient. The patient needs to have an e-mail address, an electronic device (smartphone, tablet or computer) and means for electronic identification in order to receive the intervention. As an intervention this e-therapy is available for patients both in primary and secondary care. For this trial we enroll only patients who have not received the intervention previously.

In addition to the study intervention, patients receive treatment as usual (TAU).

**Treatment as usual (TAU).**   The patients have an initial appointment with a physician and may in addition have appointments with a physiotherapist, an occupational therapist, a psychologist and/or a social worker. If needed, the patient can be in contact with the physician remotely (by phone or message in a patient portal) or at a control appointment. Patients may also attend group interventions. If possible, medication is kept the same during the intervention and three months afterwards. If there is need to change the medication during the trial, the changes will be recorded. We will also record the use of other simultaneous interventions (such as e.g. psychotherapy). Appointments with health professionals and attendance in group interventions are documented in the electronic patient record system.

The control group consists of treatment as usual for six months, which is the primary end point of the study. After this, control patients are offered the possibility to take part in either the AIR program or the HUS internet therapy if they so wish.

## Outcome assessment

The primary outcomes in the study are self-reported functional ability and quality of life as measured on a scale from 0 to 10 and functional ability measured by the WHO Disability

**Table 1. Outcomes of the RCT and the timepoints of measurement.**

| | Baseline, 0 months | 3 months | 6 months | 12 months |
|---|---|---|---|---|
| **Primary outcome measures** | | | | |
| Functional ability 0–10 | x | x | **x** | x |
| Quality of life 0–10 | x | x | **x** | x |
| Functional ability: WHODAS2.0 | x | x | **x** | x |
| **Secondary outcome measures** | | | | |
| Baseline information: other chronic diseases, medications, social status, occupation | x | | | |
| Follow-up questionnaire (PREMs, treatments, feedback) | | x | x | x |
| Quality of life: EuroHIS-QOL-8, [50] | x | x | x | x |
| Health related Quality of life: 15D [51] | x | x | x | x |
| Symptoms, SSD-12 [52] and PHQ-15 [53] | x | x | x | x |
| Depression, PHQ-9 [53] | x | x | x | x |
| Anxiety, GAD-7 [54] | x | x | x | x |
| ISI, Insomnia Severity Index [55] | x | x | x | x |
| Work ability (subjective, three steps) [56] | x | x | x | x |
| Resilience | x | x | x | x |
| Resilience Scale-14 (RS-14) [57] | | | | |
| Researcher collects information from patient health record: diagnoses, medications, healthcare use, and other possible treatments during 12 months | | | | x |

Assessment Schedule (WHODAS) instrument at six months. All outcomes of the study are presented in Table 1.

In addition to the collected patient reported outcome measures (PROM), at 3, 6 and 12 months, we will collect information on patient reported experience (PREM) of the treatment received and of participating in the AIR trial.

We will perform a modifier analysis on baseline characteristics that may influence outcome impact.

## Statistical analysis

A research assistant will store the baseline data in pseudonymized format into the HUSeCRF database. Access to the database is given to those members of the research group who analyze the data. Access to HUSeCRF is with personal codes and includes phone verification.

The codes that link patient information and study numbers are saved in the Key register, which can only be accessed by the PI and the study coordinators.

## Analysis plan

All randomized participants will be analyzed in accordance with the intention-to-treat principles. Baseline data will be summarized as counts and percentages for categorical variables and mean values with standard deviation (SD) or median values with interquartile ranges (IQRs) for normally distributed or skewed continuous variables, respectively. Changes in primary and secondary outcomes over time and differences between intervention groups will be examined using generalized linear mixed models with time (0, 3, 6 and 12 months), intervention group, and interaction between group and time as fixed effects, and participants as a random effect. The most suitable correlation structure will be determined from the data based on multiple measures, including Akaike's and Bayesian Information Criteria. Causal Directed Acyclic Graphs will be used in order to select a minimal adjustment set of fixed effect variables for

mixed modeling on repeated measures. Benjamini–Hochberg procedure will be applied to correct for multiple testing. The statistical analyses will be carried out using R statistical computing environment [58].

## Power calculations

It is hypothesized that the AIR program is equal in effect to the e-therapy. Additionally, it is hypothesized that both the AIR program and the e-therapy are more effective than TAU. According to simulations carried out based on a previous report [25] it was estimated that a sample size of at least 90 individuals per experimental group is required for reaching statistical power of >80% for outcome measures demonstrating improvement in AIR patients as compared to treatment equivalent to relaxation therapy (RT). This sample size was sufficient for all 9 out of 9 previously studied outcome measures reflecting functional impact, clinical severity, pain catastrophizing, severity of anxiety and depressive symptoms, perceived health status, psychological inflexibility, and mindfulness at post-treatment and three-month follow up. To overcome loss to follow up, we aim at 120 patients per each treatment arm. Please see details on the power analyses in S2 Appendix.

## Ethics and dissemination

This study is conducted in voluntary health care units with voluntary patients. For patients, it means an opportunity to receive an additional online program and possibly get help for the symptoms they suffer from. Participating in the study means filling in forms, which can be time consuming. In addition to the online program, the trial does not change the treatment as usual the patients receive. Helsinki University Hospital Ethics Board has approved the study plan, please see supporting information file S2 File Ethical Statement (HUS22392021) and file S1 File Protocol Approved by Ethics Committee (English). The trial is registered at ClinicalTrials.gov (Trial registration number NCT05212467). The trial is still ongoing. The study findings will be reported in a peer-reviewed journal. Any important protocol modifications will result in written amendment to the protocol and ethics application and will also be communicated to relevant parties.

## Discussion

To our knowledge this trial is the first large-scale randomized controlled trial that will explore the interventions developed to overcome the central sensitization of the brain and assess them through a patient centered approach. The results will provide valuable information on the management of conditions with persistent somatic symptoms. Good quality RCTs are necessary in determining the effectiveness of non-drug based online interventions that are available and easily accessible to patients both in primary and secondary care. Currently the resources assigned for treating persistent somatic symptoms are insufficient, and many patients are out of the reach of adequate treatment. Online self-management programs have been developed for the treatment of many other conditions [59] and could offer a treatment option for at least some of the patients with symptoms of central sensitization.

The patients attending the study are secondary care patients. They will have been admitted to the outpatient clinics of HUS based on a referral evaluated and accepted by a specialist. Approximately 50% of primary care referrals are accepted to the secondary clinics in this study and they supposedly represent the most severe cases of patients suffering from persistent somatic symptoms. Being accepted to the outpatient clinics of HUS indicates that other medical reasons to the patient's symptoms have most likely been excluded.

Within the studied patient groups there might be a difference in the willingness of the patients to participate in the study with the ones experiencing most debilitating symptoms not feeling able to commit to self-management options. From a health service provision point of view, providing easily accessible therapies for those who feel capable of following online programs might allow for the more efficient allocation of resources also for those patients, who need more support and attention. Conversely, it might also be that the patients with the most severe symptoms are the most motivated to try any possible treatment in the hope to relieve their symptoms. Patients may also have very different histories and current life situations. The effects of these factors will be tackled by randomization to obtain comparable patient groups in the study arms.

The results of the study will primarily be generalizable to other selected secondary care patients suffering from severe forms of persistent somatic symptoms. The offered treatment as usual will be received by all participating patients and it might also influence the perceived effectiveness of the interventions. An important part of the treatment relies on the patients being truly heard by the professional(s), validating their feelings, and giving them psychoeducation on their condition. The lack or inadequacy of this part of the treatment in primary health care in general (due to i.e., lacking resources or education on the conditions and their treatment) might have a significant role on the patient reported outcomes of self-management programs. Educating primary healthcare professionals working with patients suffering from persistent somatic symptoms.

on the principles and practical performance of the treatment of patients with persistent somatic symptoms might be necessary.

In this study the efficacy of the interventions will only be evaluated by subjective patient reported outcome measures. The diagnosis of persistent somatic symptoms is based upon reported subjective symptoms. Consequently, subjective symptoms are the most valid endpoints, and aiming to improve these symptoms are considered treatments [60]. Some, but not all, of the used PROMs have been validated with patients suffering from one or some of the conditions examined in this study. Quality of life and functional ability measured on a scale from 0 to 10 are susceptible to many subjective factors. On the other hand, the treatment of persistent somatic symptoms aims at improving the patient's quality of life and functional ability most importantly from the patient's subjective viewpoint. The systematic collection of patient-reported outcomes using several different surveys will give us indication not only of the effectiveness of the interventions but also allow for the evaluation of the sensitivity and reliability of the different PROMs in the assessment of effectiveness in treating persistent somatic symptoms. Furthermore, the investigated interventions accentuate the patients' active role in the treatment of their persistent somatic symptoms and may enhance patient enablement.

The interventions require access to and the ability to use an electronic device (smartphone, tablet or pc) as well as an e-mail address and means for electronic identification. For a small number of patients these requirements might hinder their participation.

Some patients might drop out if they find the interventions too intensive to follow. We will aim at minimizing dropout rates by the proactive approach of the study nurse. Having enough patients and randomizing them to the different study arms aims at reducing the effect of possible baseline confounding factors. Defining persistent somatic symptoms is challenging as these disorders lack objective findings. We plan to assess effects of the interventions in the predefined sub-groups of fibromyalgia, ME/CFS, and post COVID-19 patients. If the effects differ in these groups, we may need to continue recruitment and aim at 120 patients per treatment arm in the subgroup.

If the two clinics cannot recruit enough patients, we plan to recruit other centers, including primary care practices. Then 'treatment as usual' might mean different things for the different

units and their patients and would have to be carefully documented in terms of health service use, content of services and participating professionals. Connected to the study, participating centers will receive an introduction on persistent somatic symptoms, the underlying mechanisms, and treatment opportunities. Thus, the study will act as means for educating and disseminating valuable knowledge on persistent somatic symptoms and post COVID -19 condition to healthcare professionals. This will benefit the patients, also in the longer term. Implementing systematic use of patient reported outcome measures will also introduce new, increasingly relevant, and applicable value-based healthcare approaches to both professionals and patients.

## Conclusion

This randomized trial assesses two online methods developed to overcome the central sensitization of the brain through a patient centered approach. The results may have an impact on the management of these common and often debilitating conditions. In addition, online programs may improve patient enablement and allow for savings in costs and more effective allocation of resources in the treatment of persistent somatic symptoms and post COVID-19 condition. If the hypothesis of the study is confirmed, further research, including cost effectiveness analyses of the novel treatment models, would bring valuable information regarding the scalability, applicability, and cost-benefits of the self-treatment programs.

## Supporting information

**S1 Checklist. SPIRIT 2013 checklist.**
(DOCX)

**S1 Appendix. Diagnostic criteria.**
(DOCX)

**S2 Appendix. Power analyses.**
(PDF)

**S1 File. Protocol approved by ethics committee (English).**
(PDF)

**S2 File. Ethical statement (HUS22392021).**
(PDF)

## Author Contributions

**Conceptualization:** Helena Liira, Markku Sainio, Antti Malmivaara, Jan-Henry Stenberg, Jari Arokoski.

**Data curation:** Mikko Varonen.

**Funding acquisition:** Helena Liira, Mari Kanerva.

**Methodology:** Helena Liira, Mikko Venäläinen.

**Project administration:** Helena Liira, Mikko Varonen.

**Supervision:** Helena Liira.

**Writing – original draft:** Velina Vangelova-Korpinen, Helena Liira, Samu N. Kurki, Antti Malmivaara, Mikko Venäläinen.

**Writing – review & editing:** Helena Liira, Markku Sainio, Antti Malmivaara, Mari Kanerva, Jan-Henry Stenberg, Mikko Venäläinen, Piia Vuorela, Jari Arokoski.

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
