## [Decision Letter · Decision Letter 0]

1 Oct 2024

PONE-D-23-36707Effectiveness of mindfulness-based online therapy or internet-delivered cognitive behavioral therapy compared to treatment as usual among patients with persistent somatic symptoms: Protocol for a randomized controlled trial

PLOS ONE

Dear Dr. Vangelova-Korpinen,

Thank you for submitting your manuscript to PLOS ONE. After careful consideration, we feel that it has merit but does not fully meet PLOS ONE’s publication criteria as it currently stands. Therefore, we invite you to submit a revised version of the manuscript that addresses the points raised during the review process.

We look forward to receiving your revised manuscript.

Kind regards,

Dahua Yu

Academic Editor

PLOS ONE

“Funding

This work is supported by European Union's Horizon Europe research and innovation programme (Long Covid HEU Grant no. 101057553 2022-2026) and State Research Funds. The funding sources have no other role in the study.”

“This work is supported by European Union's Horizon Europe research and innovation programme (Long Covid HEU Grant no. 101057553 2022-2026) and State REsearch Funds. The funding sources have no other role in the study.”

4. Please ensure that you refer to Figure 1 in your text as, if accepted, production will need this reference to link the reader to the figure.

Reviewers' comments:

Reviewer's Responses to Questions

**Comments to the Author**

1. Does the manuscript provide a valid rationale for the proposed study, with clearly identified and justified research questions?

Reviewer #1: Yes

Reviewer #2: Yes

2. Is the protocol technically sound and planned in a manner that will lead to a meaningful outcome and allow testing the stated hypotheses?

Reviewer #1: Yes

Reviewer #2: Partly

3. Is the methodology feasible and described in sufficient detail to allow the work to be replicable?

Reviewer #1: Yes

Reviewer #2: No

4. Have the authors described where all data underlying the findings will be made available when the study is complete?

Reviewer #1: No

Reviewer #2: No

5. Is the manuscript presented in an intelligible fashion and written in standard English?

Reviewer #1: Yes

Reviewer #2: No

6. Review Comments to the Author

You may also provide optional suggestions and comments to authors that they might find helpful in planning their study.

Reviewer #1: In this study protocol, a three-arm randomized control trial is being proposed which aims to investigate the effectiveness of a mindfulness-based amygdala and insula retraining online program, an internet delivered therapist assisted therapy in addition to standard therapy, and standard treatment alone. Functional ability and quality of life surveys will be collected from participants at baseline 3, 6, and 12 months. Total target accrual is 360.

Minor revisions:

1- Line 277: Consider replacing “numbers” with “counts”.

2- Line 281: Indicate the type of underlying covariance structure that will be used in the linear mixed model or the criteria for selecting it.

3- Identify the software that will be used for statistical analysis.

Reviewer #2: The study accessed the efficacy of online interventions on overcoming the central sensitization of the brain in patients with persistent somatic symptoms. They found that the online programs may improve patients’ enablement.

Overall, the manuscript deals with an interest research topic. The current version of manuscript can be improved by considering the comments below.

1.The authors should illustrate the central sensitization of the brain and how the online interventions overcome central sensitization in detail.

2.The study included 360 participants. Dose participant heterogeneity have an impact on the results. For example, age/gender?

3.In the section of the Statistical Analysis, the specific statistical method should be provided.

4.The framework of the paper needs to be readjusted. The discussion part is the fifth part, but there are no 1-4 in the manuscript. Additionally, the paper lack results section.

7. PLOS authors have the option to publish the peer review history of their article (what does this mean?). If published, this will include your full peer review and any attached files.

Reviewer #1: No

Reviewer #2: No

---

## [Author Response · Author response to Decision Letter 0]

13 Nov 2024

Dear Editor, Reviewer#1 and Reviewer 2,

Thank you for your valuable comments. We have made a concerted effort to adequately respond to each comment and suggestion received from the reviewers. We hope that our efforts have improved the manuscript to meet your high standards and make it acceptable for publication in PLOS ONE. We have provided a separate file ('Response to reviewers') with point-by-point responses in a table. Below we provide the same comments in order.

EDITOR COMMENT:

RESPONSE:

Thank you for this kind note. 

We have checked the style guide and made corrections to the following naming of files/captions. 

Fig 1. SPIRIT schedule of enrollment, interventions and assessments

Fig 2. Study design

S1. File. SPIRIT 2013 Checklist

S2 Appendix. Diagnostic criteria

S2 Appendix. Power Analysis

S4 Protocol Approved by Ethics Committee (English)

S5 Ethical Statement (HUS22392021)

EDITOR COMMENT:

“This work is supported by European Union's Horizon Europe research and innovation programme (Long Covid HEU Grant no. 101057553 2022-2026) and State REsearch Funds. The funding sources have no other role in the study.”

RESPONSE:

Thank you for noticing this. We have removed funding information from the manuscript and mention it now in the cover letter: 

‘This work is supported by European Union's Horizon Europe Research and Innovation Programme (Long Covid, EU Grant no. 101057553 2022-2026) and State Research Funds. The funding sources have no other role in the study.’

Please ensure that it will be mentioned in the Funding Statement section.

EDITOR COMMENT:

When completing the data availability statement of the submission form, you indicated that you will make your data available on acceptance. We strongly recommend all authors decide on a data sharing plan before acceptance, as the process can be lengthy and hold up publication timelines. Please note that, though access restrictions are acceptable now, your entire data will need to be made freely accessible if your manuscript is accepted for publication. This policy applies to all data except where public deposition would breach compliance with the protocol approved by your research ethics board. If you are unable to adhere to our open data policy, please kindly revise your statement to explain your reasoning and we will seek the editor's input on an exemption. Please be assured that, once you have provided your new statement, the assessment of your exemption will not hold up the peer review process.

RESPONSE:

Thank you for pointing out these details. 

Below is our revised data availability statement: 

Data sharing is not applicable to this article as no datasets were generated or analyzed during the current study. 

EDITOR COMMENT:

Please ensure that you refer to Figure 1 in your text as, if accepted, production will need this reference to link the reader to the figure.

RESPONSE:Thank you for noticing this. Please see addition on line 185. 

EDITOR COMMENT:

Please include captions for your Supporting Information files at the end of your manuscript, and update any in-text citations to match accordingly 

RESPONSE:

Please see amendments for the first comment above. We have now included captions for our supporting information at the end of the manuscript and updated our in-text citations.

REVIEWER COMMENTS

REVIEWER#1

COMMENT:

1- Line 277: Consider replacing “numbers” with “counts”. 

RESPONSE: 

Thank you for this suggestion. We have changed the wording. Please see line 299

COMMENT:

2- Line 281: Indicate the type of underlying covariance structure that will be used in the linear mixed model or the criteria for selecting it 

RESPONSE:

Thank you for helping us clarify and update our analysis plan. 

The most suitable correlation structure will be determined from the data based on multiple measures, including Akaike’s and Bayesian Information Criteria. This is now included in our updated data analysis plan, please see lines 301-308.

COMMENT:

3- Identify the software that will be used for statistical analysis. 

RESPONSE:

The statistical analyses will be carried out using R statistical computing environment. Please see lines 309-310.

REVIEWER#2

COMMENT:

The authors should illustrate the central sensitization of the brain and how the online interventions overcome central sensitization in detail 

RESPONSE:

Thank you for this comment and the opportunity to further clarify these entities. 

Please see the following additions: 

lines 67-71: These predisposing, triggering and maintaining factors lead to structural, functional and chemical changes in the central nervous system causing alterations in processing of internal and external stimuli, i.e. central sensitization. The mechanisms of central sensitization have been widely studied among patients with chronic pain

lines 144-145: Meditation training has been shown to decrease the reactivity of the amygdala to emotional stimuli also during non-meditative states.

lines 150-152: Overall, there is strong evidence of the structural and functional changes in the brain following mindfulness meditation in numerous conditions, including chronic pain, a widely studied condition of central sensitization. 

COMMENT:

The study included 360 participants. Dose participant heterogeneity have an impact on the results. For example, age/gender? 

REPONSE:

Thank you for the possibility to clarify the manuscript on this topic. Please see additions made on lines 287-288:

We will perform a modifier analysis on baseline characteristics that may influence outcome impact.

The Discussion already states the following on lines 391-393: 

Having enough patients and randomizing them to the different study arms aims at reducing the effect of possible baseline confounding factors. 

COMMENT:

The framework of the paper needs to be readjusted. The discussion part is the fifth part, but there are no 1-4 in the manuscript. 

RESPONSE:

Thank you for noticing this error. We have removed ‘5.’ from the ‘Discussion’ section. 

COMMENT:

Additionally, the paper lack results section. 

RESPONSE:

Please see line 332-334: 

We have added the following on line 332: ‘The trial is still ongoing.’

---

## [Decision Letter · Decision Letter 1]

8 Dec 2024

Effectiveness of mindfulness-based online therapy or internet-delivered cognitive behavioral therapy compared to treatment as usual among patients with persistent somatic symptoms: Protocol for a randomized controlled trial

PONE-D-23-36707R1

Dear Dr. Vangelova-Korpinen,

We’re pleased to inform you that your manuscript has been judged scientifically suitable for publication and will be formally accepted for publication once it meets all outstanding technical requirements.

Kind regards,

Dahua Yu

Academic Editor

PLOS ONE

Additional Editor Comments (optional):

Reviewers' comments:

Reviewer's Responses to Questions

**Comments to the Author**

1. Does the manuscript provide a valid rationale for the proposed study, with clearly identified and justified research questions?

Reviewer #1: Yes

Reviewer #2: Yes

2. Is the protocol technically sound and planned in a manner that will lead to a meaningful outcome and allow testing the stated hypotheses?

Reviewer #1: Yes

Reviewer #2: Yes

3. Is the methodology feasible and described in sufficient detail to allow the work to be replicable?

Reviewer #1: Yes

Reviewer #2: Yes

4. Have the authors described where all data underlying the findings will be made available when the study is complete?

Reviewer #1: Yes

Reviewer #2: Yes

5. Is the manuscript presented in an intelligible fashion and written in standard English?

Reviewer #1: Yes

Reviewer #2: Yes

6. Review Comments to the Author

You may also provide optional suggestions and comments to authors that they might find helpful in planning their study.

Reviewer #1: All comments have been adequately addressed.

Reviewer #2: I have no suggestions for the revised manuscript, and the current version of manuscript can be accepted.

7. PLOS authors have the option to publish the peer review history of their article (what does this mean?). If published, this will include your full peer review and any attached files.

Reviewer #1: No

Reviewer #2: No

---

## [Editor Report · Acceptance letter]

17 Dec 2024

PONE-D-23-36707R1 

PLOS ONE

Dear Dr. Vangelova-Korpinen, 

I'm pleased to inform you that your manuscript has been deemed suitable for publication in PLOS ONE. Congratulations! Your manuscript is now being handed over to our production team.

Kind regards, 

on behalf of

Prof. Dahua Yu 

Academic Editor

PLOS ONE